# Historical Silk: A Novel Method to Evaluate Degumming with Non-Invasive Infrared Spectroscopy and Spectral Deconvolution

**DOI:** 10.3390/ma16051819

**Published:** 2023-02-22

**Authors:** Ludovico Geminiani, Francesco Paolo Campione, Carmen Canevali, Cristina Corti, Barbara Giussani, Giulia Gorla, Moira Luraschi, Sandro Recchia, Laura Rampazzi

**Affiliations:** 1Dipartimento di Scienza e Alta Tecnologia, Università degli Studi dell’Insubria, Via Valleggio 11, 22100 Como, Italy; 2Centro Speciale di Scienze e Simbolica dei Beni Culturali, Università degli Studi dell’Insubria, Via Sant’Abbondio 12, 22100 Como, Italy; 3Dipartimento di Scienze Umane e dell’Innovazione per il Territorio, Università degli Studi dell’Insubria, Via Sant’Abbondio 12, 22100 Como, Italy; 4Museo delle Culture, Villa Malpensata, Riva Antonio Caccia 5, 6900 Lugano, Switzerland; 5Dipartimento di Scienza dei Materiali, Università di Milano-Bicocca, Via Roberto Cozzi 55, 20125 Milan, Italy; 6Istituto per le Scienze del Patrimonio Culturale, Consiglio Nazionale delle Ricerche (ISPC-CNR), Via Cozzi 53, 20125 Milano, Italy

**Keywords:** samurai armor, silk, sericin, FTIR spectroscopy, spectral deconvolution, PCA

## Abstract

To correctly manage a collection of historical silks, it is important to detect if the yarn has been originally subjected to degumming. This process is generally applied to eliminate sericin; the obtained fiber is named soft silk, in contrast with hard silk which is unprocessed. The distinction between hard and soft silk gives both historical information and useful indications for informed conservation. With this aim, 32 samples of silk textiles from traditional Japanese samurai armors (15th–20th century) were characterized in a non-invasive way. ATR-FTIR spectroscopy has been previously used to detect hard silk, but data interpretation is challenging. To overcome this difficulty, an innovative analytical protocol based on external reflection FTIR (ER-FTIR) spectroscopy was employed, coupled with spectral deconvolution and multivariate data analysis. The ER-FTIR technique is rapid, portable, and widely employed in the cultural heritage field, but rarely applied to the study of textiles. The ER-FTIR band assignment for silk was discussed for the first time. Then, the evaluation of the OH stretching signals allowed for a reliable distinction between hard and soft silk. Such an innovative point of view, which exploits a “weakness” of FTIR spectroscopy—the strong absorption from water molecules—to indirectly obtain the results, can have industrial applications too.

## 1. Introduction

Preventive conservation in museums is essential to carry cultural heritage from the past to future generations. To correctly manage different materials, an assessment about their nature should be scheduled, by means of micro-invasive and non-invasive analytical techniques [1,2]. Moreover, chemical analyses for conservation purposes often offer the chance to deepen the knowledge about a collection, thus enhancing its historical value. Silk textiles are a good example of the issue, as few scientific works deeply investigated silk variability and decay despite the wide geographical and historical diffusion of this precious yarn. Silk textiles have been appreciated for their strength, luster, and vivid color which is obtained through the dyeing process.

Silk is obtained from the *Bombyx mori* silkworm. The silk cocoon is treated with hot water to obtain the single filaments which are successively wound together to give a raw silk thread or *grège* [3]. Then, the raw silk generally undergoes the degumming process, where hot water and other substances are employed to remove the gum covering [3,4,5,6]. Gum is mainly composed of sericin, which is soluble in hot water, unlike the two core brins of fibroin. More details are given in the Appendix B. According to the degree of degumming of the silk fiber, three different yarns could be distinguished.

(i)Raw or hard silk. It is obtained after twisting some single filaments together in order to obtain the thread. The sericin coating is still present, which makes dyeing difficult [3].(ii)Partially degummed or supple silk [3].(iii)Degummed or soft silk. It is the most diffused material, which possesses the typical lustrousness and smoothness. The fiber can be dyed easily [3,7].

The practice of partially or totally degumming silk is related to the geographical, historical, and cultural context which the silk was intended for [3]. There are few recent examples of hard silks in collections, such as the United States first ladies’ gowns at the Smithsonian Institution [8], but it is probable that many other collections of silk textiles could be made of supple silk. On the contrary, hard silk can easily be found in the archaeological contexts of East-Asian countries, on condition that they are found in arid burial environments that are essential to preserving ancient fibers with an intact sericin coating [6]. In China [3,6], during the first millennium C.E., silk with gum was often found as the ground cloth for paintings and writings, due to its stiffness. Instead, soft silk was present in a profusion of garments and banners, often resist-dyed or embroidered in order to create a polychrome design. For a long time, in Japan [9], only samurai and rich merchants could afford silk (*kinu*), which had been imported already degummed from China and Korea since the third century C.E. Since the Edo period, it started to be produced locally, but it remained for a long time as a luxury product, which was reserved for the well-off, as sumptuary laws prevented peasants from using this yarn. They could use only raw silk and production waste, which were spun into a coarse and matt yarn. High-quality silk was extensively used in samurai armors to make the *odoshi-ge* (lacing among metal plates) and for linings brocades. It is reported that hard silk was chosen for the samurai armor, as the stronger fiber improved armor quality in battle [10]. Soft silk has been diffused in armor making since the Edo period for the sake of aesthetics to obtain more vivid colors and a more lustrous effect. In that period, the armor had in fact lost its practical value and became a ceremonial dress.

The presence of the remnant sericin gum may have a consequence on the preservation of silk artefacts. In 1995, Becker [8,11] first proposed that even if sericin yellows, it can protect silk fibroin against light-induced damage. The role of sericin as a free-radical-scavenging antioxidant was confirmed recently [12]. In 2011, Zhang and co-workers [6,13] tested mock-up samples under heat and moisture and found that sericin can provide some protection against fibroin deterioration, but in high humidity environments, soft and hard silk ages at the same rate because of the leaching of sericin. For this reason, the most ancient samples, which could be made of hard silk, are hardly likely to retain their sericin coating. Since heat and moisture are important factors which may be relevant in promoting the decay of silk within historic collections, any wet treatment with aqueous solvents is detrimental to the preservation of the sericin coatings [6,8], as well as too high humidity conditions during exposure or conservation could be harmful [6,14]. Specific studies devoted to raw silk suggest that it should be stored below 50% RH. As a matter of fact, silk is known to be a very hygroscopic textile fiber, with hard silk being even more hygroscopic [14]. The reason for this behavior could be related to silk’s amino acidic composition (more details in the Appendix B). Hard silk contains both fibroin and sericin, while soft silk is composed of fibroin only [15]. Sericin contains three times the polar side groups of fibroin. The higher content of polar side groups makes sericin more prone to bind water molecules and soluble in hot water [7]. As an example, under standard atmospheric conditions, i.e., 27 °C and 65% RH (relative humidity), mulberry raw silk fiber has a moisture regain of 11% *w*/*w*, which reduces to about 9% *w*/*w* after degumming [16]. Zhang et al. report that both hard and soft silk show a reduction of water sorption related to ageing [6]. Another issue which affects water uptake is crystallinity. While sericin has an amorphous structure [17], fibroin is characterized to show both crystalline and amorphous domains [15]. The superior mechanical properties in terms of toughness which are attributed to silk fiber depend on this particular mixed structure so any variation in the size of crystallites and in the crystallinity degree could affect the physical properties of the yarn. The amorphous region is responsible for the elasticity, while highly ordered regions play a major role in determining the strength and stiffness. As happens with other natural polymers, water acts as a plasticizer in fibroin, penetrating only the amorphous regions’ fiber [7]. Therefore, when silk is stored in an environment with a relative humidity below 40% or at high temperatures, it can desiccate, becoming rigid, brittle and less soft.

Most of the scientific investigations about silk focus on fibroin, and only a few studies deal with sericin. Many of them investigate sericin as a biomaterial and deal with its extraction [4,5,6,17,18]. The ageing behavior of hard silk has been theoretically studied using mock-up samples [6,8,11,13,19]. To the best of our knowledge, only a few authors were interested in detecting sericin on historical silk [8,11,19,20]. Initially, amino acid analysis was used to detect sericin [8,11,19]. Zhang et al. first proposed the use of FTIR spectroscopy [13,18], even if they admitted that amino acid analysis showed the best performance [6]. Traditional transmission FTIR spectroscopy has proved to be a sensitive technique for the identification of natural fibers and in particular of silk fibroin, since the first studies in the Fifties [21,22]. However, ATR and reflection modes are definitely easier and more rapid to use, even if some modification of spectra can appear [23,24]. ATR-FTIR spectroscopy is commonly used to characterize fibroin [25,26,27,28] and especially its secondary structure [2,13,25,29], but pure sericin has also been investigated [30,31]. The possibility to use microsamples and no need for pretreatment makes the technique widely used for the study of cultural heritage materials [32,33,34,35,36]. External reflection FTIR (ER-FTIR) spectroscopy is a reflection technique as well and uses an extended MIR (medium infrared) region, collecting signals from 7500 to 375 cm^−1^. It has demonstrated great possibilities in the last decade, thanks to its portability and non-invasiveness; it has been tested mainly in the study of mortars [37] and pigments [38,39,40], but it has also been confirmed to be a sensitive technique to detect silk fibroin and sericin [18,41,42]. Nevertheless, some problems with the interpretations of spectra can occur, in the form of bands’ distortions and variations of their intensity ratio, due to the influence of both the physical and optical properties of the surface that is being investigated [24]. Thus, it is often difficult to make direct comparisons between peaks in ATR and external reflection modes, requiring the construction of dedicated databases [41]. An advantage of the extended spectral range is that a part of the NIR (near infrared) region is available for interpretation. Pure NIR spectroscopy in reflection mode has also been used to study silk, using chemometrics for the interpretation. Firstly, NIR spectroscopy can assess the nature of the yarn [43,44,45]. Secondly, some theoretical studies used the technique to indirectly estimate silk decay by measuring the loss of tensile strength [46] and the different moisture sorption of aged silk [47]. However, the last method cannot be applied to museum collections, due to the need to conditionate the textile under different fixed RHs, which is not possible in museum spaces and is problematic for the conservation of artefacts. On the contrary, NIR spectroscopy is a well-known method to measure dampness in the textile industry [45]. Finally, Mossotti et al. [48] first used NIR data associated with chemometrics to detect sericin on textiles, considering water as an interferent for the analyses.

### Aim of the Work

External reflection FTIR spectroscopy is proposed in our work as a simple and non-invasive technique to discriminate hard and soft silk samples from a collection of historical silk. The present work arises as a part of a challenging work of characterization of the ancient materials of Japanese traditional armors belonging to Museo delle Culture (Lugano, Switzerland). The project is of particular interest, as this kind of applied art has never been tested so thoroughly and extensively before. ER-FTIR spectroscopy was chosen to analyze historical samples mainly due to its non-invasiveness. This permitted us to investigate a great number of samples, chosen to represent the majority of colors and textures. As ER-FTIR spectroscopy is a practically new technique for silk recognition, the band assignment has been discussed and compared with the well-studied ATR mode, focusing also on part of the NIR spectral range, which is not commonly considered. Studies about water uptake in silk have been evaluated and reviewed in accordance with hard silk features and with data from water absorption tests. Instead of considering water as an interferent for the analysis, the signals arising from water in the O-H stretching band were studied to detect differences in the affinity to the water of hard and soft silk. Peak fitting analysis was used to confirm and quantify the differences in the spectra. Finally, principal component analysis was applied to propose visual discrimination of the two kinds of silk.

This work demonstrates for the first time that ER-FTIR spectroscopy is a successful tool to differentiate hard and soft silk, in historical samples too. The recognition of hard silk textiles provides doubly valuable information. Besides its historical significance, the detection of sericin is essential for preventive conservation and for targeted restoration works.

## 2. Materials and Methods

### 2.1. Reference Silk

Modern samples of hard and soft *Bombyx mori* silk were used as reference materials. Some of them were obtained from Museo della Seta (Como, Italy), while others were borrowed from Centro Tessile Serico Sostenibile (Como, Italy). The reference samples are the following:

-HS1 and HS2: Hard silk cloth-HS3: Hard silk yarn-SS1, SS2, and SS3: Soft silk cloth

Water absorption by the reference silk samples was controlled using two different strategies. High humidity conditions were reproduced by storing the samples for 65 h in a desiccator over K_2_SO_4_ saturated aqueous solution. The relative humidity level was constantly monitored using a data logger and set at approximately 97%. The dry condition was reproduced by leaving the samples for 65 h at 40 °C in a ventilated laboratory oven. The samples were generally analyzed under ambient laboratory conditions, except where explicitly stated in the text.

### 2.2. The Morigi Collection of Traditional Japanese Armors

The analyses discussed in this paper were carried out on a representative core of the collection of Japanese samurai armors which Museo delle Culture in Lugano received from collector Paolo Morigi in 2017. After a limited exhibit in 2018, the works of art became part of the museum’s permanent collections. The armors cover a wide range of styles and historical periods, from the Azuchi–Momoyama era (second half of the 16th century) to the Showa era (1926 to 1989) through to the Edo period (1603–1868), as reported in Table 1 and Appendix A.

Some armors were actually used in battle, as can be inferred from the fit that was comfortable and protected; others were exhibited only for celebrations and parades. Further details on the eras to which they belong have been described elsewhere [32].

The armors which are the object of this study and the sampling points are shown in Table 1. The armors silk textiles showed great variability in their colors and typology. Details for each analyzed sample are reported in Appendix A. The parts of the traditional Japanese armor are shown in Appendix A. Non-invasive analyses with ER-FTIR spectroscopy were performed on different areas which were chosen to represent the majority of the colors and textures.

### 2.3. Attenuated Total Reflectance Fourier Transform Infrared Spectroscopy

ATR-FTIR spectra were acquired with a Thermo Scientific Nicolet iS10 instrument equipped with a fast recovery deuterated triglycine sulphate (DTGS) detector. The parameters used were 32 scans, 4 cm^−1^ resolution, and a range between 4000 and 600 cm^−1^. A background spectrum was acquired periodically to allow the software to automatically subtract the atmospheric air spectrum from that of the sample. The spectra obtained are generally presented omitting the 2400–1800 cm^−1^ region, which is not very informative because it shows, at around 2100 cm^−1^, the typical absorption of the crystal of analysis consisting of a diamond. The ATR-FTIR analysis was conducted on the reference samples.

### 2.4. External Reflection Fourier Transform Infrared Spectroscopy

In situ analyses on the Morigi collection were performed in ER-FTIR mode, using a portable Alpha Bruker FTIR spectrophotometer equipped with an external reflection module for contactless measurements and a DTGS detector. The analysis parameters used are 200 scans, 4 cm^−1^ resolution, and a range of 7500–375 cm^−1^. Periodically, a background spectrum was acquired using a flat gold mirror. The measurement area was approximately 6 mm in diameter and the instrument was placed in a frontal position relative to the analysis point, at a working distance of approximately 1–1.5 cm. Fine-tuning of the optimal distance was then achieved by searching for the maximum signal directly in the interferogram using the software. The acquired spectra were processed using pseudo-absorbance [log (1/R); R = reflectance] as the intensity unit.

### 2.5. Data Treatment and Elaboration

The spectra of the samples were interpreted by comparing them with the reference samples and the literature. The optical spectroscopy software Spectragryph, version 1.2.16.1, was used to visualize and manipulate the ATR-FTIR and ER-FTIR spectra [49]. The same software was used to convert native ER-FTIR spectra into .jdx files in order to manipulate them with chemometrics.

Baseline correction was applied to all spectra. FTIR spectra are also commonly pre-processed to remove the effects of light scattering phenomena. For this purpose, the SNV (standard normal variate) method is often used to effectively remove multiplicative interference of scattering and particle size [50]. This pre-processing method eliminates the information about the absolute intensity of the signals but enhances the subtle differences in the band shape of the different superimposed spectra. In this study, SNV pre-processing was applied, when necessary, to the entire spectrum or only to a region of it.

The application of SNV is based on the following mathematical operation:(1)$$y_{ij\;\left(\mathrm{SNV}\right)}=\frac{y_{ij}-\bar{y}}{\sqrt{\frac{\Sigma {\left(y_{i}-\bar{y}\right)}^{2}}{n-1}}}$$

That is subtracting the mean spectra $\bar{y}$ to each intensity value $y_{i}$ of the original spectrum and then dividing for the standard deviation value.

### 2.6. Spectral Deconvolution/Curve-Fitting Analysis

Based on the literature and our previous work, the OH stretching band was analyzed using a band fitting method [28,32,51,52]. The selected spectra were cut to the range of 3800–2400 cm^−1^; then, a baseline correction was applied using a linear function passing through the ordinates at the endpoints of this interval, and SNV correction was performed too. The FitPeaks Pro function of the peak analyzer package of Origin Pro 2018 software (OriginLab Corporation) was used for band fitting as follows.

As a first step, the second derivative of the convoluted spectra was calculated and smoothed using the adjacency averaging method (measurement smoothing window 20). This made it possible to identify the position of the bands, which were then compared with the literature. Next, the spectra were deconvoluted using Gaussian curves and a constant baseline (constrained to zero absorbance). Some bands were allowed to shift from their initial position, within a specific range, while the full width at half height (FWH) of the bands was fit to a specific range based on the theoretical width of the band [53]. Table 2 shows the bounds setting. The fitting was iterated until convergence and a Chi-Sqr tolerance value of 10^−6^ was obtained.

A similar method was developed in order to deconvolute the water band at 5170 cm^−1^ in the NIR region. The spectral region between 5400 and 5000 cm^−1^ was selected, smoothed (Savitsky–Golay method, interval = 21, polynomial order = 2), and baselined. The position of the bands was found by means of the second derivative, accordingly to the literature [47]. The bands were assigned as follows: non-freezing water, 5050 cm^−1^; freezing bound water, 5140 cm^−1^; bulk water, 5220 cm^−1^.

### 2.7. Principal Component Analysis

ER-FTIR datasets were subjected to principal component analysis (PCA). All data were centered before further analysis. Prior to model calculation, different preprocessing techniques were tested and evaluated in order to correct unwanted data modification such as, as an example, different scattering. The preprocessing step was optimized by assessing the suitable mathematical transformation to remove the unwanted artefacts from the spectra. The Savitsky–Golay derivative, SNV, baseline correction, range reduction, and a combination of them were tested. The software used for the chemometric calculations was R version 3.6.3 (Rstudio version 1.4.1106).

## 3. Results and Discussion

Firstly, the ATR-FTIR and ER-FTIR spectra of soft silk are reported and compared, as a complete band assignation for silk fibroin with ER-FTIR has never been discussed before. The spectral differences arising from water uptake are then evaluated for both ATR-FTIR and ER-FTIR spectra, by means of water absorption tests. Finally, reference hard silk is investigated, and compared with soft silk in order to find a key for discrimination. Peak fitting analysis is used to validate our supposition. At the end, the proposed method is tested on a case study, by applying it on historical silk samples. PCA is applied to visually detect samples made of hard silk.

### 3.1. ATR-FTIR and ER-FTIR Spectra of Soft Silk

The band assignment for the ATR-FTIR spectra has been discussed and published extensively for fibroin [18,25,29]. The main absorption bands are due to the absorptions by amides A, B, I, II, and III, which are typical for the protein backbone [53,54]; alongside these absorptions which are shared by all proteins with little variations, other signals arise from the amino acids’ side chains, such as ν(CC) and δ(CH) in tyrosine, ν(C=O) in aspartic acid, and ν(CO) in serine [53,54,55,56]. Figure 1 compares the spectra of the same reference of soft silk taken with ATR-FTIR and ER-FTIR spectroscopy.

The spectra appear very different; in particular, some shifts appeared mainly in amides A, I and II’s peaks. At first sight, the peaks at 1706 cm^−1^ and 1680 cm^−1^ appear extremely enhanced by external reflection, while below 1450 cm^−1^, no sensitive differences are noticed. Amides I and II’d peaks apparently show a great shift. In our opinion, their intensities were probably enhanced to the point that they appear as inverted peaks. This is a common problem with the ER-FTIR mode [40], but also with diffuse reflectance infrared Fourier transform spectroscopy (DRIFT) [27]. Thus, some peaks should be considered as inverted and their maxima could be identified at 1698, 1628–1618, and 1508 cm^−1^. The assignment of ER-FTIR bands is proposed in Table 3, making a tentative comparison with ATR-FTIR spectral features which are reported in the literature.

### 3.2. Water Uptake Behavior of Soft Silk

Another characteristic of the ER-FTIR spectrum (Figure 1) is the broadening of the band at 3330 cm^−1^, which creates a single band together with the water OH stretching band between 3400 and 3600 cm^−1^, as previously reported [18]. The enhancement of the -OH signal with respect to the ATR spectra is typical of the ER-FTIR mode [41].

The spectral region between 3600 and 3000 cm^−1^ is generally associated with intramolecular and intermolecular hydrogen bonding and free hydroxyls in polar macromolecules, such as cellulose, but also with the free or the bound water linked to the substrate. Water FTIR signals are strongly influenced by their state of aggregation [59]. In particular, water molecules bind in different forms when they are adsorbed on a protein film [54,64] or a biocompatible polymer [57,58], such as silk [7]. On the interface, water and the C=O and N-H groups of the protein backbone form hydrogen bonds, some of which can replace direct N-H⋯O=C hydrogen bonds which are typical of crystalline domains of fibroin. This water is the so-called non-freezing water, as it never crystallizes due to the tight bond to carbonyl groups. Freezing bound water, which instead crystallizes below 0 °C, interacts with the other polar groups in the side chain. Finally, bulk or freezing water, which crystallizes at ∼0 °C, has a bulk-water-like structure with an O-H---O-H hydrogen bond network and it weakly adsorbs to the surface. As the degree of hydrogen bonding between water and protein increases, the FTIR peaks are shifted to higher wavenumbers [54]. Assignments for each type of O-H stretching are reported in Table 3 and compared with the same vibrations in the ATR mode. Some shifts between the ER and ATR modes were experienced. Some soft silk samples were conditioned under different RH conditions to evaluate spectral differences which arise from the water uptake of silk. The spectra were taken both with ATR-FTIR mode and the ER-FTIR mode, and they are shown in Figure 2.

As expected, soft silk shows very hygroscopic behavior. The bands which are associated with water uptake are highlighted in Figure 2a,b. Under increasing RH conditions, the overall intensity of the spectra increases, thus suggesting the enhancement of the broad bulk water absorption at around 3220 cm^−1^. Similarly, both signals associated respectively to freezing bound (3420–3400 cm^−1^) and non-freezing water (3560–3500 cm^−1^) are strongly influenced by both low and high humidity conditions. Their intensity is enhanced, and the spectral shape is changed. No shifts appear yet. When humidity conditions are changed, the ATR-FTIR and ER-FTIR modes show different responses. The ATR-FTIR spectrum at low humidity appears very different from the spectra under ambient and high humidity conditions, while the ER-FTIR spectra under low and ambient conditions are similar. In the ER-FTIR spectra, it is interesting to also note the OH combination band at 5170 cm^−1^, whose intensity is strongly enhanced only under high humidity conditions. In Figure 2b, the different contributions to the overall band are indicated and assigned to the different types of water. These contributions are discussed in Section 3.3. As the conditioned samples were analyzed with ATR-FTIR and ER-FTIR paying particular attention to maintaining the correct conditioning, the results suggest that ER-FTIR is not so strongly influenced by low humidity conditions with respect to the ATR mode.

### 3.3. Characterization of Hard and Soft Silk

The possibility to discriminate hard and soft silk with ATR-FTIR spectroscopy was the theoretical basis of this research project. Band assignment for the ATR-FTIR spectra has been published extensively for fibroin [18,25,29]. Sericin shows similar signals [18,30,31] and the main source of the slight differences lies in their distinct conformation of the secondary structure, in addition to differences in amino acid composition [25,65]. The literature [17,18,30,31] reports that slight shifts in amides I and II’s peaks are the main signs of sericin’s presence, together with a broader amide A band at 3270 cm^−1^ and a water sorption band centered at 3400 cm^−1^, which are signals for raised hydroxyl content. Generally, the authors report that peaks at about 1400 cm^−1^ (C-H and O-H bending [17,25,31]) and 1075–52 cm^−1^ (C-OH stretching [25,31] or C-C bending [54,65]) are distinctive of serine [18,30,31,54] and as a consequence are useful to distinguish sericin (mainly composed of serine) from fibroin [20,65]. Moreover, Zhang et al. [18] suggest that a decrease in intensity of the 1000 and 975 cm^−1^ peaks, which are typical of fibroin, could infer the presence of a sericin coating. Generally speaking, it appears that the distinction between hard and soft silk is challenging, as there are no evident peak shifts or spectral features belonging to hard silk only. It is important to also consider the possibility of sericin leaching due to high humidity conditions [18].

The reference samples of hard and soft silk were analyzed with ATR-FTIR spectroscopy to evaluate the best markers for differentiating hard and soft silk. The spectra of reference samples of hard and soft silk (Figure 3) confirm what is reported in the literature. In particular, the decreased intensity of peaks at 3270 and 1440 cm^−1^ and the increased intensity at 2920, 2850, and 1071 cm^−1^ could be markers for hard silk detection. The decrease in the peaks at 1000 and 975 cm^−1^ is quite difficult to notice. In our opinion, the best indicator is the broadening of the bands at 3500–3420 cm^−1^. Such broad bands can be attributed to hydrogen-bonded water, whose absorption can be found between 3600 and 2900 cm^−1^ according to the strength of hydrogen bonding [54,66]. The shoulder at 3500 cm^−1^ is ascribable to H-bonded OH to C=O of the amide [67]. Another interesting difference is the increase in the band at 3220 cm^−1^. Both silk types have a high capability to adsorb moisture, but hard silk is more prone to bind water due to its composition and amorphous structure, as discussed in the Introduction.

External reflection infrared spectroscopy was applied on the same reference materials to test if the same spectral features which characterize hard silk in the ATR-FTIR spectra could be recognized. Appendix A show the ER-FTIR spectra for all of the references of hard and soft silk. The instrumental spectral range is split into two spectra (range 6100–3800 cm^−1^ and range 3800–400 cm^−1^). The region 7500–6100 cm^−1^ of the ER-FTIR spectrum is not shown. For clarity, only two references are shown in Figure 4, which represent the range 3800–900 cm^−1^.

In this region, the main differences are located in the band of the hydrogen-bonded water and in the intensities of some of amides I and II’s peaks. Assuming that the maxima of amides I and II are inverted peaks, we point out the shift of the absorption from 1618 (hard silk) to 1626 cm^−1^ (soft silk) and the differences between the samples and references at around 1510 cm^−1^. The inverted peak at 1618 cm^−1^ attests the higher content of β-strands for hard silk, while the shoulder at 1650 cm^−1^ is a sign of the high content of random coil conformation. Both findings agree with the description of hard silk. Similarly, a decrease in intensity is observed at 1680 cm^−1^, probably due to β-turn content which is lower in hard silk. The amide II peak at 1560 cm^−1^ shows a shift and a decrease in intensity, too. At around 1510 cm^−1^, the inverted peaks are due to C-N and N-H from amide II. Another characteristic of the hard silk spectrum is the impressive broadening of the band between 3400 and 3600 cm^−1^, which is more enhanced with respect to soft silk and centered around the new peak at 3560 cm^−1^. The variations in intensities which are distinctive for hard silk in the ATR mode are not present, so other markers for sericin should be identified.

We think that the different water uptake values under the same environmental conditions are the key to discrimination. The amorphous fraction of silk (sericin and the disordered fraction of fibroin) is mainly affected by the absorption of non-freezing water, as crystalline regions are hardly accessible to water [47,68]. As a matter of fact, the amorphous phase is responsible for 70% of the uptake of environmental water [69]. The amino acid composition could have some influence too, as the polar amino acid serine is the main component of sericin with 30% *w*/*w* in contrast to the serine content in fibroin which is nearly 15% *w*/*w* [18]. In particular, where the structure presents grooves, as in the case of hard silk, the nature of the side chain has greater importance in the interactions with water molecules [70]. Therefore, the combined influence of the morphology and the amino acid composition could make hard silk more prone to bind water with respect to soft silk under the same environmental conditions. These considerations are visualized in Figure 5.

In order to study the different contributions of water to the overall band, a peak fitting analysis was carried out. Figure 6a,b shows the deconvolution models of the region of 3800–2400 cm^−1^. Some areas under the Gaussian curve forming the overall curve are reported in form of percentages to highlight the difference between the two kinds of silk (Figure 6c).

In hard silk, the area of the water band centered at 3560 cm^−1^ is definitely higher than in soft silk. This band is related to non-freezing water, which is evidently associated to the amorphous structure which is accessible to water. The band at around 3630 cm^−1^, which is assigned to non-hydrogen bonded water, is higher in hard silk than in soft silk, while the signal of freezing bound water (around 3420 cm^−1^) is unexpectedly higher in soft silk, but this band can also be influenced by the contribution of the amide A peak. In the ATR mode, the signal is located at 3270 cm^−1^, but the literature [54] reports that H-bonding could be responsible for a blueshift towards 3310 cm^−1^. Actually, both are probably present, as N-H groups exist in two forms, both C=O-bonded (NH---OC) and water-bonded (NH---OH_2_), giving rise to two different signals [71]. At 3270 cm^−1^, the absorption is due to intermolecular bonded N-H stretching, as confirmed by theoretical calculations [72]; at 3313 cm^−1^, the signal could be ascribable to water-bonded N-H stretching, as it is experimentally seen in the polyamides’ spectra [61,73]. Manas et al. [74] confirmed that amides show different signals at the same time, when solvent-exposed domains differ from the bulk which has little interaction with water. The blueshift is due to the NH---OH_2_ hydrogen bond, which tends to increase the force constant of the NH in-plane bending motion. Actually, it is important to recall that the region 3400–3200 cm^−1^ is overlapped with O-H stretching signals, so that it is difficult to clearly discriminate the contributions. Similarly, the signal of bulk water at 3220 cm^−1^ can be partially overlapped with N-H stretching. Anyway, it is definitely higher in hard silk than in soft silk, as we can expect as it is reported that hard silk absorbs much more water than soft silk [16].

The NIR range (7000–4000 cm^−1^) of the spectrum of hard and soft silk references HS1 and SS1 is shown in Figure 7, while Appendix A shows the same range for all of the references. The literature [47,48,75,76,77] about silk reports overtone and combination bands arising from OH (water and serine), NH (peptides), and CH (peptides, alanine, serine, etc.). The main assignments are described in Table 4. Significant changes in intensities occur in this region; while the water band at 5170 cm^−1^ (O-H combination) increases in hard silk, amides bands at 4850, 4620, and 4520 cm^−1^ (hydrogen bonded N-H and C=O vibration) are more prominent in soft silk, as experienced by Mossotti et al. [48]. Mo et al.’s study [76] showed that NIR bands are sensitive to conformational changes, so higher intensities in soft silk are probably due to the dominant conformation of fibroin, which is β-sheet. Hard silk is covered by amorphous sericin, so the signal from crystalline fibroin is lower. On the other hand, sericin makes hard silk more prone to adsorb water, probably causing enhancement of the band.

If the band at 5170 cm^−1^ is deconvoluted (Figure 8), it reveals other important information about the water degree of association with silk [47]. Due to its asymmetric aspect, it can be deconvolved into at least three components (5219, 5139, and 5046 cm^−1^). As the degree of C=O and N-H groups’ hydrogen bonding to water increases, peaks are shifted to lower wavenumbers. There are three types of signals, which could be attributed to non-freezing, freezing bound, and bulk water [76]. In all cases, water is more prone to bind hard silk, which contains an amorphous sericin covering and less β-sheet structured fibroin (the degumming process increases the crystallinity index of silk), hence the higher areas of hard silk shown in Figure 8c, obtained through the peak fitting analysis. The two higher wavenumber bands offer a distinction of bound water with different degrees of hydrogen bonding. They are located at 5222 and 5141 cm^−1^, assigned to bulk and bound-freezing water, respectively. The band at 5050 cm^−1^ is assigned to strongly hydrogen-bonded structural water, which is non-freezing water. Values associated with this band account in hard silk for more than twice as much as for soft silk. This demonstrates that the capacity to strongly bond water is higher in hard silk than in soft silk due to structural differences. The percentages for bound-freezing water are significant, too; as higher values are associated with aminoacidic composition richer in polar side chains, hard silk is confirmed to be more prone to bond freezing bound water. As for bulk water, differences in absorption are still present, even less evidently, because hydrated proteins offer a similar surface to bind bulk water. The different contributions of the combination band of water are shown in Figure 5.

### 3.4. Characterization of Silk in Traditional Japanese Armors

External reflection FTIR spectroscopy was applied to a wide selection of textiles from armors in order to test extensively the variability in the appearance and color of silk yarns. The same main peaks (1706, 1566, 1454, 1265, and 1071 cm^−1^) are found in all the spectra, so we can infer that they are all made of pure silk, without peculiar samples showing other recognizable signals. However, samples 4_2 and 4_11, belonging to the Mor.004 armor (17th century), appear different from the others, especially in their water sorption band (3200–3600 cm^−1^), which is shown in Figure 9a, and in the amides region (1800–1500 cm^−1^), which is not shown. In order to obtain Figure 9a, the spectra were truncated between 3800–2600 cm^−1^, linearly baselined, and SNV corrected. It is clear that samples 4_2 and 4_11 are different from all of the other samples, mainly due to the shape of the OH stretching band. Through comparison of this feature, the spectra of hard and soft silk samples could be visually differentiated. For the sake of clarity, in Figure 9b, just two representative spectra of the samples are shown together with the references. The spectrum of sample 4_2 strictly resembles the spectrum of the hard silk reference, and the same happens for sample 5_1 and the soft silk reference.

In Figure 2a,b, we showed how OH stretching bands in soft silk spectra are influenced under different RH conditions. It could be argued that it is not possible to distinguish between the enhanced absorption due to high humidity conditions from that caused by the presence of the sericin coating. Actually, under the same temperature and RH, hard silk is doomed to show higher absorptions in the OH stretching band with respect to soft silk. Thus, the comparison method works provided that the textiles to be compared belong to the same collection or are stored under the same RH conditions. Within the analysis of a part of a collection, it could be useful to conditionate two reference clothes—made of hard and soft silk—under the same temperature and RH conditions of the collection. Recording their spectra together with other samples would make it easier to visually evaluate the spectra. As shown in Appendix A and Figure 9b, the reference materials do not differ significantly among them and from historical hard silk samples. With respect to an earlier proposed method [48], there is no need to conditionate the textiles of the collection under fixed RH before the analysis, which would be difficult to obtain and potentially dangerous for their preservation.

Finally, it is worth pointing out that the recognition of hard silk was achieved on samples dating back to 17th century. It is a high-value consideration as it shows that the proposed method works on both modern mock-up samples and historical samples. It is proof that the decay condition does not generally prevent the identification of hard silk, provided that sericin has not been degraded. Indeed, the chance to detect sericin decreases as decay takes place. Thus, the most ancient samples, which have a higher probability of being made of hard silk for historical reasons, can hardly show spectral features of sericin. In this case, more sophisticated techniques should be used. Moreover, the proposed method cannot falsely indicate the presence of sericin in a degummed textile. Indeed, the rise of water sorption cannot be related to the aging of the samples, as both hard and soft silk show a reduction of water sorption related to aging [6].

Instead of visual comparison, principal component analysis (PCA) can be applied to the ER-FTIR spectra. The method is more rigorous and objective to evaluate differences among sets of samples and to search groups among them. Similar samples locate themselves in the same region of the scores plot, while samples belonging to different groups are far apart. PCA is an unsupervised learning algorithm, with it being able to find some patterns and regularities without direct supervision of an operator and thus objectively. The scores plot permits obtaining visual recognition of such differences. According to the purpose of this research, hard and soft silk samples should create two different groups. Samples appearing as disturbing or unusual are named as outliers, and care of them must be taken to obtain reliable models [78]. In general, the spectrum of the sample could be intended as an outlier if it lies outside the distribution obtained from those of the other samples, and it should be corrected or removed from the model. Outliers’ evaluation can be carried out according to the Hoteling T^2 statistic and the Q statistic, and their presence is due to a gross error producing an anomalous acquisition or the peculiar features of the sample in respect to the others (e.g., strongly IR-absorbing substances adhering to the textile).

In this work, different spectral ranges were tested to choose the most significant. The MIR region and NIR region were initially considered independently in order to evaluate the specific preprocessing method for each part. A successively extended NIR region (7400–2400 cm^−1^) was tested in order to enhance the information from the most important bands. Finally, the whole spectrum was considered and preprocessed with Savitsky–Golay smoothing (derivative order zero, second polynomial order, window width 71) and MSC. This appeared as the best choice as the sum of the explained variance for PC1 and PC2 was higher, and by observing these spectra, the artefacts were removed in an effective way. The relative scores plot of PC1 vs. PC2 is shown in Figure 10. These two components describe 69.9% and 7.4% of the total variation respectively.

In the scores plot, the samples are black colored. Most of them are located at low positive and low negative values on PC1 and PC2, thus forming a group in the middle of the graph. With respect to the main group, samples 4_2 and 4_11 appear at higher values on both PC1 and PC2. In order to have a visual evaluation of their nature, some hard and soft silk references were projected into the PCA. They are blue and red colored, respectively. As a matter of fact, the soft silk reference correctly joined the group in the middle of the graph, while hard silk references locate themselves near to samples 4_2 and 4_11. The hard silk reference HS3 is not shown as it was recognized as an outlier. Probably their spectra show similarities. In order to study the spectral features which contribute to such a scores plot, the analysis of the loadings plot of the first two PCs is needed.

The loadings for PC1 and PC2 are depicted as lines in Figure 11, together with the spectra of references for hard and soft silk. It is worthwhile to recall that high loading values (both positive and negative values) indicate important variables and thus, in this case, important peaks. The regions of the spectra where the loading values are higher are more significant, so they are highlighted.

The PC1 loadings showed a shape that resembles the one of the hard silk spectrum. This is not surprising as the first PC mainly describes the differences between the hard silk and the other silk samples. Negative high loadings values are strongly associated with peaks which are typical of soft silk (red colored bands in Figure 11), such as the amide A peak (3313 cm^−1^), β-turn (1681 cm^−1^), amide II (1560 cm^−1^), and amide III (1260 cm^−1^). The positive values at around 3650–3520 cm^−1^ could be attributed to an enhancement of the contribution due to non-bonded water and to non-freezing water (blue colored bands in Figure 11). Similarly, positive values at around 1770 cm^−1^ are associated with hard silk (blue colored band in Figure 11), as they appear in the band broadening in Figure 4 and Appendix A.

PC2 shows high values for variables (peaks) which can be associated with water uptake in the region 3580–3400 cm^−1^ (blue colored bands in Figure 11). A positive value at around 3580 cm^−1^ is associated with the vibration of non-freezing water, whose content is much higher in hard silk than in soft silk. On the contrary, the negative value at around 3305 cm^−1^ is associated with NH stretching (red colored band in Figure 11), which appears more evidently in soft silk. As for the NIR region, the positive value at around 5200 cm^−1^ could be associated with the raised water content of hard silk (blue colored band in Figure 11).

## 4. Conclusions

The proposed method to discriminate hard and soft silk is based on the different water uptake during the storage at the same temperature and RH, followed by ER-FTIR spectroscopy. This is an innovative point of view, which exploits a “weakness” of FTIR spectroscopy—the strong absorption from water molecules—in order to indirectly obtain the results. Indeed, OH stretching bands are generally considered “forbidden” regions, since the analytical information about the molecule under analysis is covered by environmental water, which is very difficult to remove from silk textiles too. Actually, we showed that these bands are useful to study adsorbed water, by means of peak fitting analysis which appeared as an interesting tool to evaluate the different contributions of the OH stretching band. In particular, we found that the contribution of non-freezing water is decisive for differentiating hard and soft silk.

When samples are stored under the same conditions, the higher water content, which causes the broad absorption at around 3600 cm^−1^, is linked to the presence of sericin and, indirectly, to the detection of hard silk. Thus, through the analysis of the shape of the OH stretching band, it is possible to differentiate hard and soft silk textiles, using a rapid and completely non-invasive technique. With respect to previous methods developed with ATR-FTIR spectroscopy, the proposed method is also easier, as only a single broad band had to be taken into consideration. The recognition of hard silk is obtained by visual comparison of spectra—two conditioned reference clothes made of hard and soft silk can be used to make identification easier. Alternatively, principal component analysis allows a more rigorous and objective comparison of the spectra, paying particular attention to the presence of outliers.

The proposed approach could infer very useful information about the nature and authenticity of historical silk textiles, thus suggesting the best conservation conditions and leading to targeted restoration works. Great campaigns of analyses can be managed, as the technique is rapid and non-invasive. The method could be useful within the industrial refining of silk too. Quality control analyses are fundamental to assure that the product achieves standard levels, but the measurement of the degumming extent of raw silk is difficult with traditional protocols. Instead, reflection infrared spectroscopy could be applied by manufacturers for continuous process control, as the working parameters are controlled and constant. The indirect measurement of degumming extent in the industrial context could be an interesting future outlook, even if further studies are needed to obtain quantitative data and chemometrics would be fundamental to manage them.

## Figures and Tables

**Figure 1 materials-16-01819-f001:**
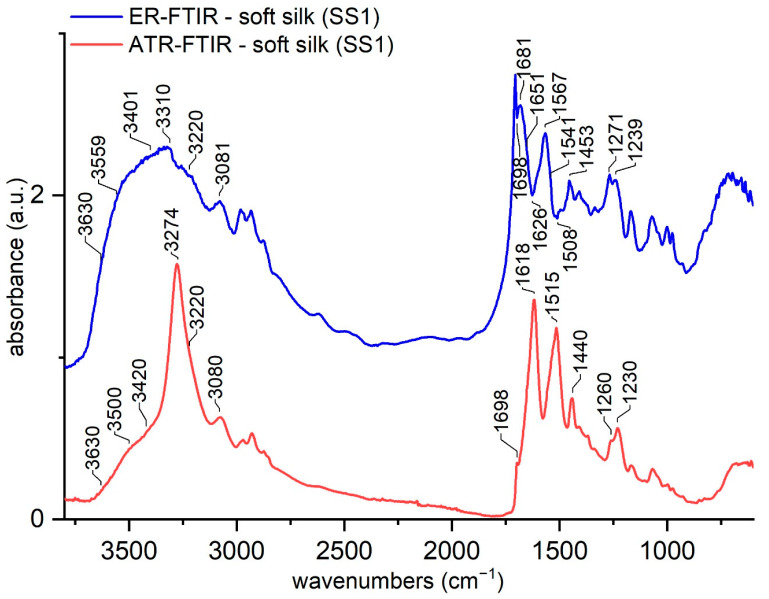
Comparison of ATR-FTIR and ER-FTIR spectra of soft silk. The region 7500–4000 cm^−1^ of the ER-FTIR spectrum is not shown.

**Figure 2 materials-16-01819-f002:**
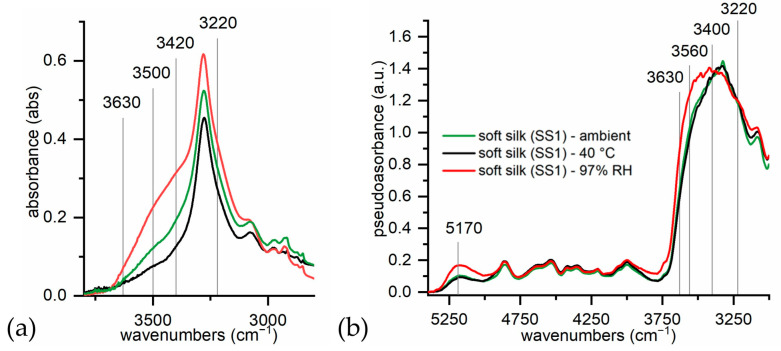
(**a**) ATR-FTIR spectra of soft silk under different relative humidity conditions; (**b**) ER-FTIR spectra of soft silk under different relative humidity conditions.

**Figure 3 materials-16-01819-f003:**
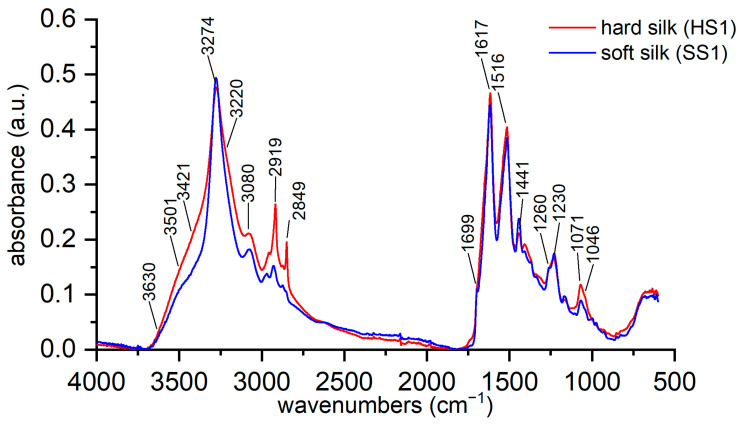
ATR-FTIR spectra of the reference samples of hard and soft silk.

**Figure 4 materials-16-01819-f004:**
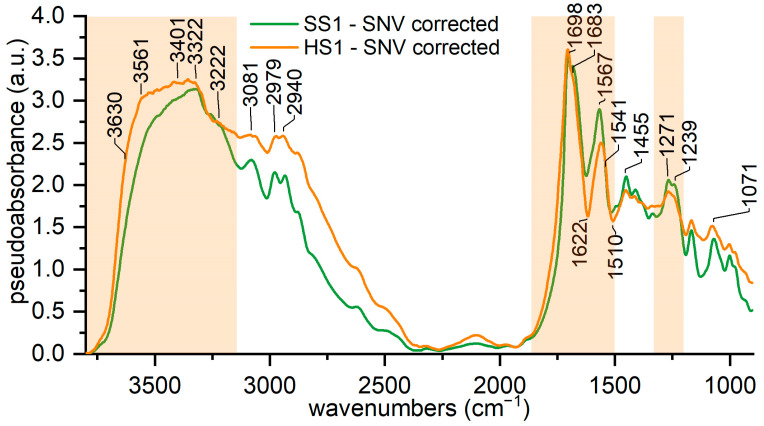
ER-FTIR spectra of hard and soft silk. The region of 3800–900 cm^−1^ is shown.

**Figure 5 materials-16-01819-f005:**
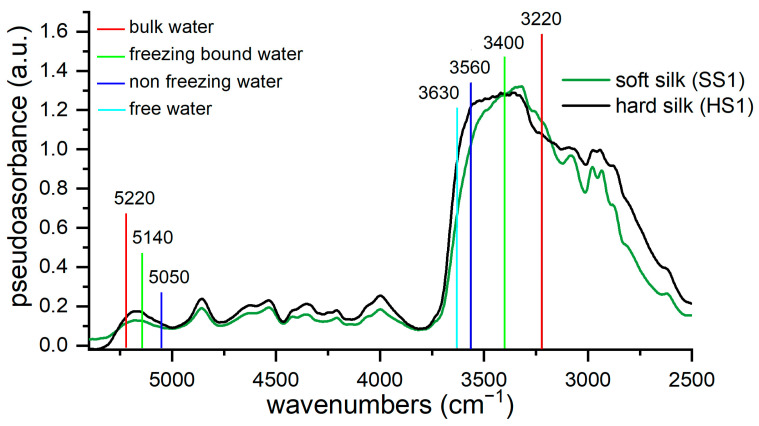
Details of the ER-FTIR spectra of hard and soft silk reference (region 5500–2250 cm^−1^). Absorptions due to different types of water are highlighted.

**Figure 6 materials-16-01819-f006:**
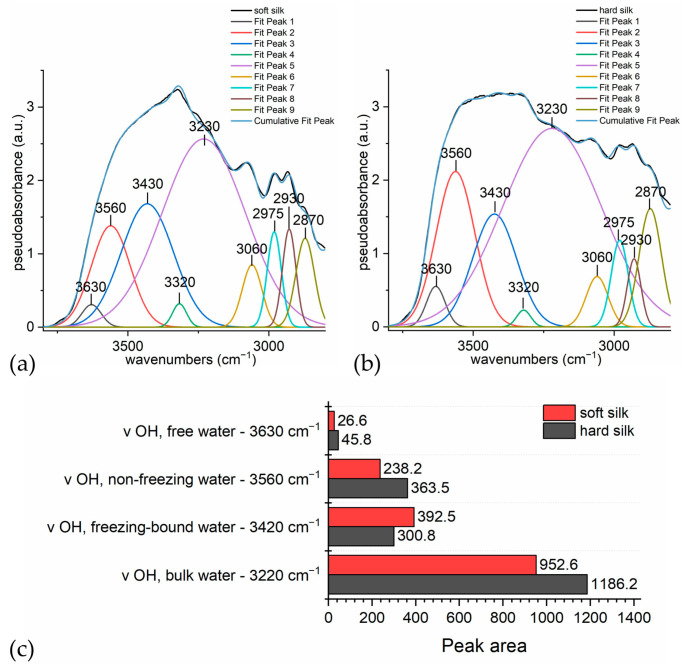
Deconvolution of the O-H stretching band (region 3715–2800 cm^−1^): (**a**) soft silk; (**b**) hard silk; (**c**) comparison of the peak areas for each contribution in soft and hard silk.

**Figure 7 materials-16-01819-f007:**
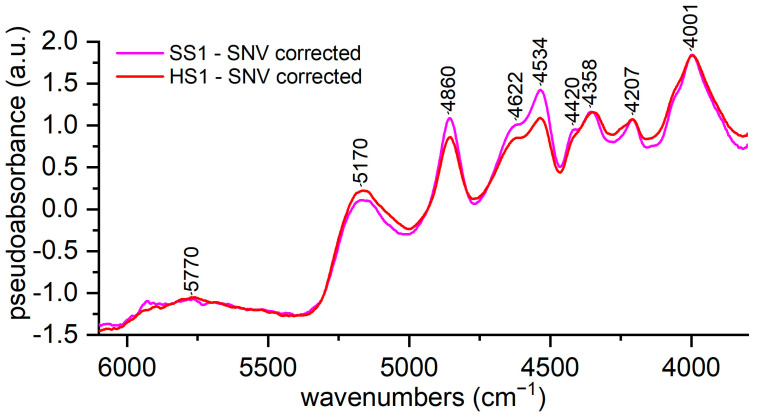
ER-FTIR spectra of hard and soft silk references. The region 6100–3800 cm^−1^ is shown.

**Figure 8 materials-16-01819-f008:**
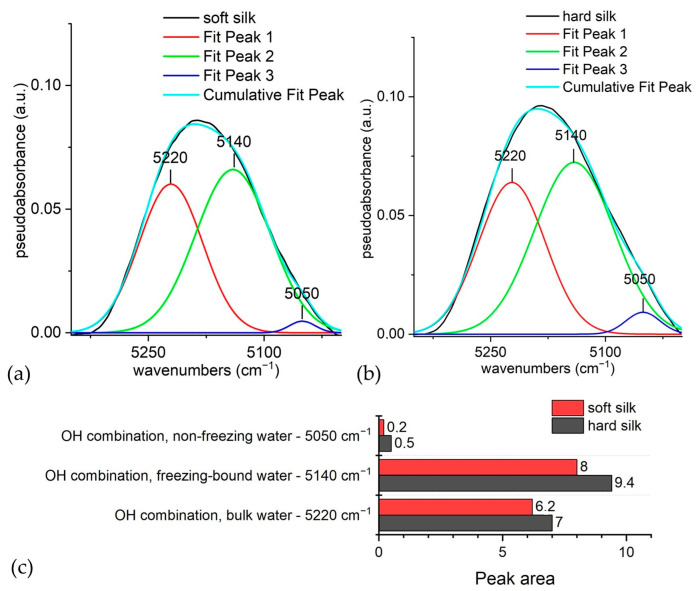
Deconvolution of the band at 5170 cm^−1^. Deconvolution of the O-H combination band (region 5350–5000 cm^−1^): (**a**) soft silk; (**b**) hard silk; (**c**) comparison of the peak areas for each contribution in soft and hard silk.

**Figure 9 materials-16-01819-f009:**
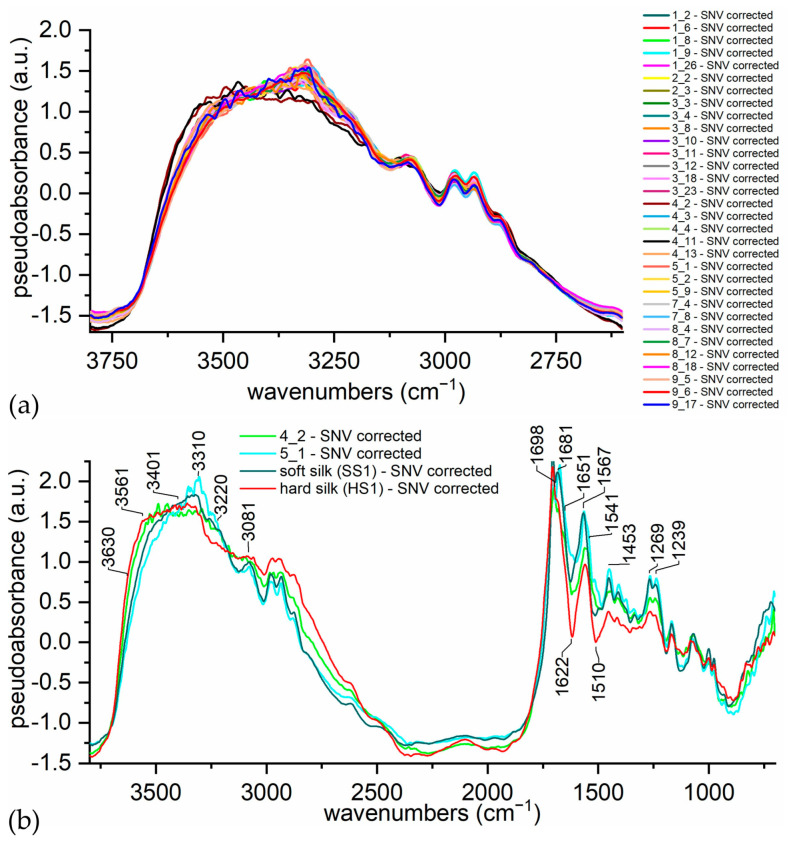
(**a**) ER-FTIR spectra of samples. (**b**) ER-FTIR spectra of samples 4_2 and 5_1 together with the spectra of hard and soft silk references.

**Figure 10 materials-16-01819-f010:**
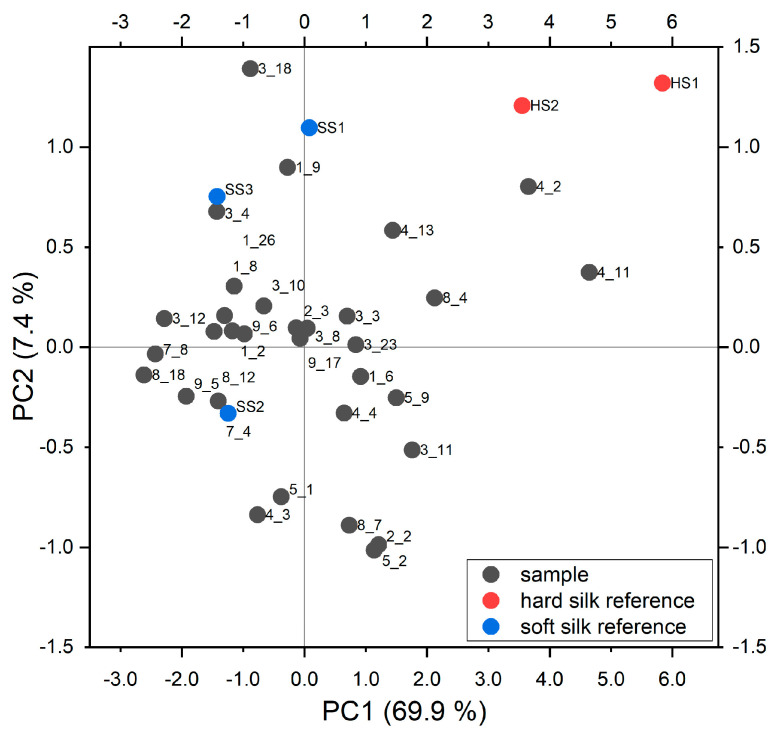
Scores plot for the first two principal components. Red and blue points are the hard and soft silk references, respectively, which were projected into the PCA.

**Figure 11 materials-16-01819-f011:**
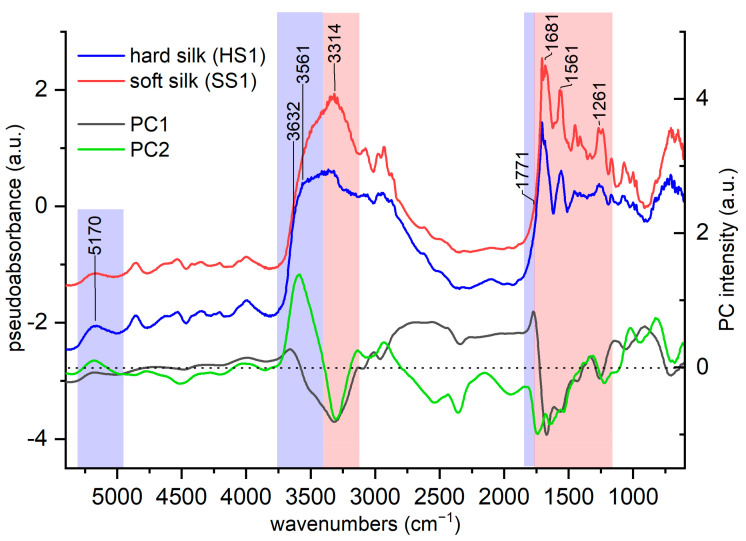
Loadings plot of PC1 and PC2, shown together with the ER-FTIR spectra of the references for hard and soft silk. The region 5400–700 cm^−1^ is shown. The blue and red regions represent the most important regions for the differentiation between hard and soft silk, respectively.

**Table 1 materials-16-01819-t001:** List of samples from the Morigi collection. A photograph of the full armor is reported with its inventory number. On the right, the samples are listed and the presumed dating for each analyzed part is reported.

**2017.Mor.1** 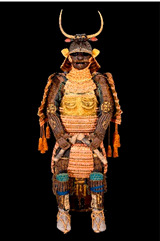	**Sample**	**Dating**	**2017.Mor.2** 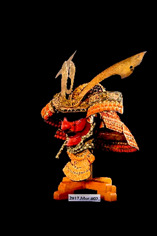	**Sample**	**Dating**
	1_2	Late 16th c.		2_2	Late 19th c.
	1_6	Late 16th c.		2_3	Late 19th c.
	1_8	Late 16th c.			
	1_9	Late 16th c.			
	1_26	Late 16th c.			



**2017.Mor.3** 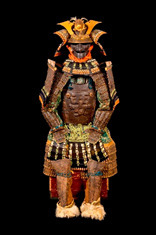	**Sample**	**Dating**	**2017.Mor.4** 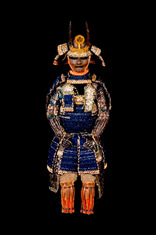	**Sample**	**Dating**
	3_3	17th c.		4_2	18th c.
	3_4	17th c.		4_3	18th c.
	3_8	17th c.		4_4	18th c.
	3_10	17th c.		4_11	18th c.
	3_11	17th c.		4_13	18th c.
	3_12	17th c.			
	3_18	17th c.			
	3_23	Late 16th c.			

**2017.Mor.5** 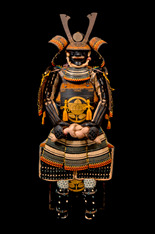	**Sample**	**Dating**	**2017.Mor.7** 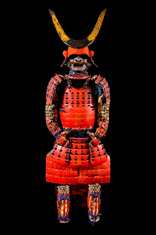	**Sample**	**Dating**
	5_1	After 1926		7_4	17th c.
	5_2	After 1926		7_8	17th c.
	5_9	After 1926			





**2017.Mor.8** 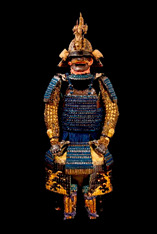	**Sample**	**Dating**	**2017.Mor.9** 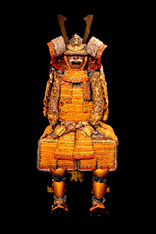	**Sample**	**Dating**
	8_4	17th c.		9_5	Late 19th c.
	8_7	17th c.		9_6	Late 19th c.
	8_12	Early 16th c.		9_17	Late 19th c.
	8_18	Early 16th c.			





c. is the abbreviation for a century.

**Table 2 materials-16-01819-t002:** Bounds setting for the curve fitting of the OH-stretching band.

Peak	1	2	3	4	5	6	7	8	9
Centre (cm^−1^)	3630	3560	3400	3320	3220	3060	2977	2931	2875
Centre bounds (cm^−1^)	±15	±15	±30	±15	±30	±5	±5	±5	±5
FWHH bounds (cm^−1^)	0–200	0–200	0–200	0–80	0–350	0–80	0–50	0–50	0–50

**Table 3 materials-16-01819-t003:** Comparison between the spectral features of the ER-FTIR and ATR-FTIR modes. The most evident shifts are in bold.

ER-FTIR (cm^−1^)	ATR-FTIR (cm^−1^)	ATR-FTIR Band Assignment
~3630	3630	Free OH (water dimers) [54]
**~3560**	**3500 (sh)**	Non-freezing water (O-H---O=C) [57,58]
**~3400**	**3420 (sh)**	Freezing bound water (O-H---polar groups) [57,58,59]
**3320**	**3274**	Amide A, N-H stretching [18,42]
3220	3220	O-H stretching, bulk water (---OH---OH---OH---) [59,60,61]
3080	3080	Amide B, N-H stretching [54]
1698 (inv)	1698 (sh)	ν(C=O) amide I bond, β-sheets [25,29]
**1682**	**1675 (sh)**	Amide I, β-turn [25,29]
1650	1650	Amide I, α-helix/random coil [25,29]
1627–17 (inv)	1628–1621	Amide I, intermolecular β-sheets [25,29]
	1621–1616	Aggregated β-strand/intermolecular β-sheet
**1567**	**1555**	Amide II, β-sheets [25,29]
1541	1545	Amide II, α-helix/random coil [25,29]
1508 (inv)	1515	Amide II, β-sheets [25,29]
1453	1440	CH_2_, CH_3_ bending in alanine [25,29]
1270	1260	Amide III, β-sheets [62,63]
1238	1230	Amide III, α-helix/random coil [62,63]

(sh): shoulder; (inv): inverted peak.

**Table 4 materials-16-01819-t004:** List of peaks and their assignations in the NIR range of the spectrum (7000–4000 cm^−1^).

Wavenumber (cm^−1^)	Assignment [47,48,75,76]
7500–6000	Water, first overtone
5900–5700	ν(CH), first overtone
5170	O-H combination
4860	Amide A ν(NH) + amide I/amide II
4620	Amide A ν(NH) + amide III/amide B + amide II
4534	Amide A ν(NH) + amide III/amide B + amide II
4420	ν + δ(CH)
4358	ν + δ(CH)
4205	ν + δ(CH)

## Data Availability

The data presented in this study are available on request from the corresponding author.

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
