# Peer review of "Historical Silk: A Novel Method to Evaluate Degumming with Non-Invasive Infrared Spectroscopy and Spectral Deconvolution"

_materials, 2023, doi:10.3390/ma16051819_

Round 1

Reviewer 1 Report

This manuscript proposed using ER-FTIR spectroscopy to to study adsorbed water, by means of peak fitting analysis which appeared an interesting tool to evaluate the different contributes of OH stretching band. This is a very interesting work and should be considered of acceptance after correcting following comments:

1.       A few sentences are too long and the information is quite confusing, please check and paraphrase them into clear messages. Such as “After degumming, silk loses up to 25% of its weight, so it generally undergoes the weighting process, in order to replace with inorganic salts or other substances some of the lost weight, or even to exceed the original weight, and to make fibres more prone to bind dyes (by adding mordanting agents)[5,7]” Also, are the references are for binding dyes or the weight loss?  Another example is: “So, besides its historical significance, the detection of sericin is essential for preventive conservation. Most of the scientific investigations about silk focuses on fibroin, and only few studies deal with sericin. Many of them investigate sericin as a biomaterial and deal with its extraction [3,4,6,15,16]. The ageing behaviour of hard silk was theoretically studied using mock up samples [6,8,11,13,17]. To the best of our knowledge, only few authors were interested to detect sericin on historical silk [8,11,17,18].” This can be combined into two sentences without over repeating the same message.

2.       The manuscript structure should be well organized. For instance, the beginning of introduction provided over much information that are not very related, instead, the “Theoretical background” could be moved to introduction.

3.       Check the references position to make a correct citation. “Such broad band can be attributed [49,62] to hydrogen bonded water, whose absorption can be found between 3600 and 2900 cm-1 according to the strength of hydrogen bonding.”

4.       Figure 6 a and b are showing the peak fitting, which is very important for this work but the fitting should be further refined to best fit the experimental data.

5.       A suggestion over the sample labelling in figures it that use sample id only and describe the ER-FTIR and ATR-FTIR in legend.

Author Response

We thank very much the reviewer for her/his words of appreciation. She/he could grasp the meaning of our work and raised some issues that we have tried to resolve, hopefully improving the quality of the article.

  1. A few sentences are too long and the information is quite confusing, please check and paraphrase them into clear messages. Such as “After degumming, silk loses up to 25% of its weight, so it generally undergoes the weighting process, in order to replace with inorganic salts or other substances some of the lost weight, or even to exceed the original weight, and to make fibres more prone to bind dyes (by adding mordanting agents)[5,7]” Also, are the references are for binding dyes or the weight loss?  Another example is: “So, besides its historical significance, the detection of sericin is essential for preventive conservation. Most of the scientific investigations about silk focuses on fibroin, and only few studies deal with sericin. Many of them investigate sericin as a biomaterial and deal with its extraction [3,4,6,15,16]. The ageing behaviour of hard silk was theoretically studied using mock up samples [6,8,11,13,17]. To the best of our knowledge, only few authors were interested to detect sericin on historical silk [8,11,17,18].” This can be combined into two sentences without over repeating the same message.

We thank the revisor for the suggestion. Sentences across the text were rephrased to make them clearer. Please refer to the revised manuscript.

  1. The manuscript structure should be well organized. For instance, the beginning of introduction provided over much information that are not very related, instead, the “Theoretical background” could be moved to introduction.

The introduction underwent a major reorganization. Similarly, a paragraph was added at the beginning of “Results and discussion” section in order to clarify the different aims of the work. Some information which is useful to deepen the topic was moved to the “Appendix” to make the “Introduction” more concise.

  1. Check the references position to make a correct citation. “Such broad band can be attributed [49,62] to hydrogen bonded water, whose absorption can be found between 3600 and 2900 cm-1 according to the strength of hydrogen bonding.”

We thank the revisor for the suggestion. Citations across the text were checked to make them clearer. Please refer to the revised manuscript.

  1. Figure 6 a and b are showing the peak fitting, which is very important for this work but the fitting should be further refined to best fit the experimental data.

The peak fitting was refined according to the revisor’s suggestion. Please refer to the revised manuscript.

  1. A suggestion over the sample labelling in figures it that use sample id only and describe the ER-FTIR and ATR-FTIR in legend.

We took the revisor’s suggestion where it was possible, as in Figure 2 and 11. Please refer to the revised manuscript.

Reviewer 2 Report

Abstract:

Line 27: What is the  meaning of bad performance.

It is necessary to mention how this study will advance the field – lines 33-34 is not enough in the context.

Introduction:

This work is very interesting. In fact, recent studies have not focused on this. However, author could shorten the introduction.

Authors should shorten the paragraph 2 and 3 – please try to focus on the key messages of this paragraph. I think, authors mainly investigating the hard and soft silk only – if so, usage of constant terminology is required throughout the paper. Why authors suddenly used a theoretical background. It should be the part before aim. Some historical information are unnecessary for the context.

Materials and methods:

Specimen provided in Table 1 should be well organized. It is difficult to follow. Authors should clearly explain the benefit and different of using ATR-FTIR and ER-FTIR.

Lines 255-259: need more elaboration.

The purpose of using principal component analysis is unclear.  

Results and Discussion:

Section 3.1 and 3.2: why authors suddenly focusing on spectra of soft silk. The paper should organize the content logically. What is the meaning of difference in band for ATR and ER – any shifts in band assignment should be explained  with scientific evidence such as why this difference occurs and what is the meaning of it.

Section 3.4: as the armors investigation are the key and novelty of this study, it should be focused on the initial part of the results and discussion. This results and discussion section should be properly organized. It was very difficult to follow the organization of the content and its purpose.

Conclusion:

Limitation and future study should be mentioned.  

Author Response

We thank very much the reviewer for her/his words of appreciation. She/he raised some issues that we have tried to resolve, hopefully improving the quality of the article

Abstract:

Line 27: What is the meaning of bad performance.

The meaning was clarified both in the introduction section (line 27) and in the text (line 369), as follows:

Line 27:

ATR-FTIR spectroscopy was previously tested to detect hard silk, but data interpretation is challenging.

Line 369:

Generally speaking, it appears that the distinction between hard and soft silk is challenging, as there are no evident peak shifts or spectral features belonging to hard silk only. It is important to consider also the possibility of sericin leaching due to high humidity conditions [18].

It is necessary to mention how this study will advance the field – lines 33-34 is not enough in the context.

The novelties of the study were emphasised in the abstract, as follows:

Line 31:

The ER-FTIR band assignment for silk was discussed for the first time. Then, the evaluation of the OH stretching signals allowed a reliable distinction between hard and soft silk. Such an innovative point of view, which exploits a “weakness” of FTIR spectroscopy – the strong absorption from water molecules – to indirectly obtain the results, can have industrial applications too.

Novelties are discussed diffusely in the Conclusion too, from line 590. Please refer to the revised manuscript.

Introduction:

This work is very interesting. In fact, recent studies have not focused on this. However, author could shorten the introduction.

Authors should shorten the paragraph 2 and 3 – please try to focus on the key messages of this paragraph. I think, authors mainly investigating the hard and soft silk only – if so, usage of constant terminology is required throughout the paper. Why authors suddenly used a theoretical background. It should be the part before aim. Some historical information is unnecessary for the context.

The Introduction underwent a major reorganization. Some information which is useful to deepen the topic was moved to the “Appendix” to make the “Introduction” more concise. Please refer to the revised manuscript.

Materials and methods:

Specimen provided in Table 1 should be well organized. It is difficult to follow.

The caption of Table 1 was clarified accordingly. Besides, in the Supporting Materials Table S1 contains supplementary information about the specimens. Please refer to the revised manuscript.

Authors should clearly explain the benefit and different of using ATR-FTIR and ER-FTIR.

In writing the “Materials and section”, we have followed the journal guidelines: “the Materials and Methods should be described with sufficient details to allow others to replicate and build on the published results”. As a consequence, we have discussed the potentialities and limits of the chosen techniques across the text, in particular in the introduction. Please refer to the revised manuscript at line 124, as follows:

However, ATR and reflection modes are definitely easier and more rapid to use, even if some modification of spectra can appear [23,24]. ATR-FTIR spectroscopy is commonly used to characterize fibroin [25–28] and specially its secondary structure [2,13,25,29], but also pure sericin was investigated [30,31]. The possibility to use microsamples and no need of pretreatment makes the technique widely used for the study of cultural heritage materials [32–36]. External reflection FTIR (ER-FTIR) spectroscopy is a reflection technique as well and uses an extended MIR (medium infrared) region, collecting signal from 7500 to 375 cm-1. It has demonstrated great possibilities in the last decade, thanks to its portability and non-invasiveness; it has been tested mainly in the study of mortars [37] and pigments [38–40], but also confirmed to be a sensitive technique to detect silk fibroin and sericin [41–43]. Nevertheless, some problems with the interpretations of spectra can occur, in the form of bands distortions and variations of their intensity ratio, due to the influence of both physical and optical properties of the surface that is investigated [24]. Thus, it is often difficult to make direct comparisons between peaks in ATR and external reflection modes, requiring the construction of dedicated databases [41].

Lines 255-259: need more elaboration.

The text was revised and the detailed procedure is given from line 248, as follows:

The OH stretching band was analysed by a band fitting method, based on previous similar works [28,32,51,52]. First of all, selected spectra were truncated down to the 3800–2400 cm−1 range and baseline correction was applied using a linear function passing through the ordinates at the endpoints of the considered interval. SNV correction was applied too. Band fitting was performed using the FitPeaks Pro function of the peak analyser package of Origin Pro 2018 software (OriginLab corporation), as follows. First, the second derivative of the convoluted spectra was smoothed by Adjacent-averaging method (smoothing window size of 20) and used to locate the position of bands, which were compared with literature. Then, the spectra were deconvoluted using Gaussian curves and a constant baseline (constrained at zero absorbance). Some bands were allowed to move in a specified range from their initial position, while the full width at half height (FWHH) of the bands was fit in a specifical range according to the theoretical band width [53]. Bounds setting are reported in Table 2. The fitting was iterated until convergence and a Chi-Sqr tolerance value of 10-6 were reached.

The purpose of using principal component analysis is unclear.  

We clarified the text and detailed the aim of the PCA in the discussion, at line 539, as follows:

Instead of the visual comparison, principal component analysis (PCA) can be applied to ER-FTIR spectra. The method is more rigorous to evaluate differences among set of samples and to search groups among them. Similar samples locate themselves in the same region of the scores plot, while samples belonging to different groups are far. The loading plot permits to obtain a visual recognition of such differences, if they exist. According to the purpose of the analysis, hard and soft silk samples should create two different groups.

Results and Discussion:

Section 3.1 and 3.2: why authors suddenly focusing on spectra of soft silk. The paper should organize the content logically. What is the meaning of difference in band for ATR and ER – any shifts in band assignment should be explained  with scientific evidence such as why this difference occurs and what is the meaning of it.

The analytical study of soft silk is fundamental as ER-FTIR spectroscopy was tested on silk nearly for the first time. As a consequence, it was necessary to compare ER-FTIR and ATR-FTIR spectra of soft silk. The differences in their spectra were discussed from line 298, as follows:

Spectra appear very different; in particular, some shifts appeared mainly in amides A, I and II peaks. At first sight, the peaks at 1706 cm-1 and 1680 cm-1 appear extremely enhanced by external reflection, while below 1450 cm-1 no sensitive differences are noticed. Amides I and II peaks apparently show a great shift. In our opinion, their intensities were probably enhanced to the point that they appear as inverted peaks. It is a common problematic with ER-FTIR mode [40], but also with diffuse reflectance infrared Fourier transform spectroscopy (DRIFT) [27].

The reason for the differences is due to influence of both physical and optical properties of the surface that is investigated (Nodari & Ricciardi, 2019), as explained from line 134 as follows:

Nevertheless, some problems with the interpretations of spectra can occur, in the form of bands distortions and variations of their intensity ratio, due to the influence of both physical and optical properties of the surface that is investigated [24]. Thus, it is often difficult to make direct comparisons between peaks in ATR and external reflection modes, requiring the construction of dedicated databases [41].    

Section 3.4: as the armors investigation are the key and novelty of this study, it should be focused on the initial part of the results and discussion. This results and discussion section should be properly organized. It was very difficult to follow the organization of the content and its purpose.

A paragraph at the beginning of “Results and discussion” part was added in order to clarify the different aims and steps of the work, as follows:

Line 279:

Firstly, ATR-FTIR and ER-FTIR spectra of soft silk are reported and compared, as a complete band assignation for silk fibroin with ER-FTIR has never discussed before. The spectral differences arising from water uptake are then evaluated for both ATR-FTIR and ER-FTIR spectra, by means of water absorption tests. Finally, reference hard silk is investigated, and compared with soft silk in order to find a key for the discrimination. Peak fitting analysis is used to validate our supposition. At the end, the proposed method is tested on a case study, by applying it on historical silk samples. PCA is applied to visually detect the samples made of hard silk.”

The investigation on armours is only a part of the work, as the characterization of reference materials was fundamental for the interpretation of the results. As we propose a method for the discrimination of hard silk in different textile collections, the identification of hard silk in Japanese armours was considered as the application of the proposed method to a case study, to assess its validity and highlight any critical issues.

Conclusion:

Limitation and future study should be mentioned.  

The Conclusion was modified accordingly, please refer to the revised manuscript.

Round 2

Reviewer 2 Report

This paper improved a lot.. Some minor comments

1. Abstract Line 35: what kind of industrial applications.

2. Line 170: how this research will advance the field of textile materials?  

3.  Line 612-615: conclusion should be to conclude the findings for general audience. What is the meaning of more rigorous.. what are the outliers? 

Author Response

We thank you the revisor for the words of appreciation. The answers to his/her questions follow.

Abstract Line 35: what kind of industrial applications.

The discussion about the industrial applications cannot be included in the abstract due to the words limit. However, the discussion was expanded in the conclusion, from line 628.

The method could be useful within the industrial refining of silk too. Quality control analyses are fundamental to assure that the product achieve standard levels, but the measurement of degumming extent of raw silk is difficult with traditional protocols. Instead, reflection infrared spectroscopy could be applied by manufacturers for a continuous process control, as the working parameters are controlled and constant. The indirect measurement of degumming extent in the industrial context could be an interesting future outlook, even if further studies are needed to obtain quantitative data and chemometrics would be fundamental to manage them. 

Line 170: how this research will advance the field of textile materials?  

This information is reported across the text. Please referer to the revised manuscript.

(line 167)

This work demonstrates for the first time that ER-FTIR spectroscopy is a successful tool to differentiate hard and soft silk, in historical samples too. The recognition of hard silk textiles is a doubly valuable information. Besides its historical significance, the detection of sericin is essential for preventive conservation and for targeted restoration works.

line 604

ER-FTIR spectroscopy. This is an innovative point of view, which exploits a “weakness” of FTIR spectroscopy – the strong absorption from water molecules – in order to indirectly obtain the results. Indeed, OH stretching bands are generally considered “forbidden” regions, since the analytical information about the molecule under analysis is covered by environmental water, which is very difficult to remove from silk textiles too. Actually, we showed that these bands are useful to study adsorbed water, by means of peak fitting analysis which appeared an interesting tool to evaluate the different contributes of OH stretching band. 

line 616

Thus, through the analysis of the shape of the OH stretching band, it is possible to differentiate hard and soft silk textiles, using a rapid and completely non-invasive technique. Respect to previous methods developed with ATR-FTIR spectroscopy, the proposed method is also easier, as only a single broad band had to be taken into consideration. 

Line 612-615: conclusion should be to conclude the findings for general audience. What is the meaning of more rigorous. what are the outliers? 

Both the ideas were clarified in revised version, as follows.

line 540:

The method is more rigorous and objective to evaluate differences among sets of samples and to search groups among them. Similar samples locate themselves in the same region of the scores plot, while samples belonging to different groups are far. PCA is a unsupervised learning algorithm, being able to find some patterns and regularities without direct supervision of an operator and thus objectively. The scores plot permits to obtain a visual recognition of such differences. According to the purpose of this research, hard and soft silk samples should create two different groups. Samples appearing as disturbing or unusual are named outliers, and care of them must be taken to obtain reliable models [79]. In general, the spectrum of the sample could be intended as outlier if it lies outside the distribution obtained from those of the other samples, and it should be corrected or removed from the model. Outliers’ evaluation van be carried out according to the Hoteling T^2 statistic and the Q statistic, and their presence is due to a gross error producing an anomalous acquisition or the peculiar features of the sample in respect to the others (e.g. strongly-IR-absorbing substances adhering to the textile).